# cd26 Knockdown Negatively Affects Porcine Parthenogenetic Preimplantation Embryo Development

**DOI:** 10.3390/ani12131662

**Published:** 2022-06-28

**Authors:** In-Sul Hwang, Joohyun Shim, Keon Bong Oh, Haesun Lee, Mi-Ryung Park

**Affiliations:** 1Animal Biotechnology Division, National Institute of Animal Science, Rural Development Administration, Wanju 55365, Jeonbuk, Korea; ih2386@cumc.columbia.edu (I.-S.H.); keonoh@korea.kr (K.B.O.); leehs1498@korea.kr (H.L.); 2Columbia Center for Translational Immunology, Department of Medicine, Columbia University Irving Medical Center, New York, NY 10032, USA; 3Department of Transgenic Animal Research, Optipharm, Inc., Cheongju 28158, Chungbuk, Korea; wings083@naver.com

**Keywords:** cd26, small interfering RNA, parthenogenetic embryos, pig

## Abstract

**Simple Summary:**

The cd26 gene is known to be significantly involved in immune responses. However, there are very few studies that have used cd26 siRNA in porcine embryos and no available information regarding its role during this pre-implantation development in in vitro. In the present study, the cd26 siRNA-injected oocytes showed significantly lower blastocyst development than control groups and we confirmed aberrant gene expression patterns of the porcine parthenogenetic embryo development. This study showed the possibility that the cd26 plays an important role in in vitro development of the porcine pre-implantation embryo.

**Abstract:**

cd26 is ubiquitously distributed in the body, particularly in the endothelial and epithelial cells, with the highest expression in the kidney, liver, and small intestine. In humans, cd26 serves as a marker for the embryo implantation phase. However, little is known about the role of cd26 in porcine pre-implantation embryo development. Here, we aimed to examine siRNA-induced cd26 downregulation in the cytoplasm of MII oocytes, to determine whether cd26 is involved in the regulation of porcine pre-implantation embryonic development. The cd26 siRNA was micro-injected into the cytoplasm of MII oocytes, which were then parthenogenetically activated electrically in a medium containing 0.3M Mannitol. Inhibition of the cd26 expression did not affect cleavage but stopped development in the blastocyst stage. Additionally, the cd26 siRNA-treated blastocysts had significantly more apoptotic cells than the untreated blastocysts. Among the 579 transcripts evaluated with transcriptome resequencing, 38 genes were differentially expressed between the treatment and control blastocysts (*p* < 0.05). Twenty-four genes were upregulated in cd26 siRNA-injected blastocysts, whereas 14 were downregulated. These genes are involved in apoptosis, accumulation of reactive oxygen species, and aberrant expression of ribosomal protein genes. Our results indicate that cd26 is required for proper porcine parthenogenetic activation during embryonic development.

## 1. Introduction

Pigs are increasingly used as a model organism in agricultural and biomedical research [1,2,3]. The generation of in vitro production (IVP) embryos is an essential step in producing pigs for this purpose, with common methods employed being in vitro maturation (IVM), in vitro fertilization, somatic cell nuclear transfer, parthenogenesis, and in vitro culture. However, the porcine IVP embryos have considerably lower developmental ability than the in vivo embryos, similarly to IVP embryos from other species that experience poor cytoplasm, polyspermy, and imperfect culture conditions [4,5,6,7,8,9].

As embryonic development is a complex event controlled by numerous regulatory networks, the underlying mechanisms must be better understood before the IVP embryos can be improved. To this end, we previously reported that cd26 downregulation decreased the development of parthenogenetically activated porcine embryos [10]. cd26, also known as DPP-4, is a 110-kDa membrane-associated peptidase originally identified in 1966 as a dipeptide naphthylamidase that hydrolyzes glycyl-prolyl-beta-naphthylamide [11]. The cd26 is expressed on the apical surfaces of epithelial and acinar cells, as well as in endothelial cells, fibroblasts, and lymphocytes [12,13,14,15]. The protein has multiple functions. For example, it acts as a serine protease, receptor, costimulatory protein, and adhesion molecule for collagen and fibronectin; it is also involved in apoptosis [16]. Moreover, cd26 is expressed on the extravillous trophoblasts (EVTs) of the placenta, and its enzymatic activation leads to EVT invasion in humans. Thus, it is an indicator of endometrium implantation, is expressed on the cell surface, and can be reduced to various biologically active peptidases in extracellular domains [17,18].

However, besides our previous research [10], only a few studies have evaluated the effect of cd26 on porcine early embryonic development. In mice, the inhibition of cd26 abrogates stress-induced abortion, whereas its overexpression enhances blastocyst adhesion and the outgrowth domain in the trophectoderm [19]. Here, to better understand the action of cd26 in porcine embryos, we identified molecules important for pre-implantation embryo development using RNA sequencing (RNA-Seq). Additionally, we assessed cd26-knockdown efficiency with small interfering (si)RNA. This approach may help to clarify the mechanisms underlying the cd26-regulated gene expression in porcine embryos.

## 2. Materials and Methods

### 2.1. Chemicals

All of the chemicals used in the present study were purchased from Sigma-Aldrich Chemicals (St. Louis, MO, USA) unless otherwise stated.

### 2.2. Experimental Design

The in vitro matured oocytes were randomly allocated to experimental groups. In experiment 1, the expression of cd26 mRNA and developmental competence in porcine pre-implantation embryos following siRNA injection and parthenogenetic activation was investigated. Then, transcriptomic analysis upon cd26 knockdown and validation of the DEGs was performed in experiment 2.

### 2.3. In Vitro Maturation

The in vitro maturation (IVM) protocol was performed, as previously described [20]. The porcine ovaries were collected from a local slaughterhouse (Nonghyup Moguchon, Gimje, Korea) and transported to the laboratory at about 30–35 °C in 0.9% saline. The antral follicles (3–6 mm in diameter) were aspirated, using an 18-gauge needle. The follicular fluid with cumulus oocyte complexes (COCs) was washed three times in TCM-199 with 0.1% (*w*/*v*) polyvinyl alcohol (PVA). Next, 100 COCs were matured in 500 µL Medium 199 (Thermo Fisher Science, Waltham, MA, USA), 0.57 mM cysteine, 10 ng/mL epidermal growth factor (EGF), 10 IU/mL follicle-stimulating hormone (FSH), 10 IU/mL luteinizing hormone (LH), and 10 % (*v*/*v*) porcine follicular fluid in a four-well dish. The COCs were cultured for 22 h with LH and FSH, and then for 22 h without hormones at 38.5 °C under 5% CO_2_. After 44 min of IVM, oocytes were treated with 0.1% hyaluronidase for 5 min before separating the cumulus cells using gentle pipetting. The extrusion of the first polar body was verified for use in microinjection experiments.

### 2.4. Microinjection of siRNA

The siRNA was designed and commercially synthesized by Sigma-Aldrich, with duplex sequences as follows: control siRNA; 5′-ACUUCGACACAUCGACUGC[dT][dT]-3′ and cd26 siRNA; 5′-UUAAGUAAUCAGUUAGAGUGU-3′. The siRNA at three concentrations (5, 10, and 20 µg) were assessed, with each experiment replicated five times. To knockdown cd26, approximately 10 pL of siRNA prepared in RNase-free H_2_O was micro-injected into the cytoplasm, using an inverted Nikon Diaphot microscope (200× magnification) and an electric microinjector (IM-400, Narishige, Tokyo, Japan). Non-silencing siRNA (Sigma-Aldrich) was injected into another group of oocytes (control siRNA), and a third group received a sham injection without siRNA (uninjected).

### 2.5. Parthenogenetic Activation and In Vitro Culture

The injected oocytes were activated electrically in a medium containing 0.3 M mannitol, 1.0 mM CaCl_2_, 0.1 mM MgCl_2_, and 0.5 mM HEPES. Between the electrodes (0.2 mm diameter), two direct current pulses (1.25 kV/cm) were applied for 30 µs at 1-s intervals using an Electro Cell Fusion Generator (Nepa Gene, Ichigawa, Chiba, Japan). The electro-activated oocytes were incubated in porcine zygote medium-3 (PZM-3) containing 7.5 µg/mL of cytochalasin B for 3 h. Next, they were washed twice and cultured in a four-well dish containing PZM-3 with 0.4% bovine serum albumin (BSA) at 38.5 °C under 5% CO_2_. On days 2 and 7, the number of cleavage and blastocyst embryos was recorded to calculate their formation rates.

### 2.6. TUNEL Analysis

To detect apoptosis, blastocysts were fixed using 4% paraformaldehyde in phosphate-buffered saline (PBS) and subjected to terminal deoxynucleotidyl transferase-mediated dUDP nick-end labeling (TUNEL) assay, using an In Situ Cell Death Detection Kit (Roche Diagnostics GmbH, Mannheim, Germany). The negative control embryos were placed in a well containing FITC label only, without the TUNEL enzyme. The blastocysts were also examined using 4,6-diamidino-2-phenylindole (DAPI) staining (Vector Laboratories, Burlingame, CA, USA), to count the total nuclei versus the apoptotic nuclei.

### 2.7. RNA Extraction, cDNA Library Construction, and RNA Sequencing

The total RNA was extracted from MII-oocyte (250 ea), two-cell (parthenogenetically activated (PA), 200 ea; siRNA, 195 ea), four-cell (PA, 151 ea; siRNA, 162 ea), eight-cell (PA, 155 ea; siRNA, 172 ea), morula (PA, 50 ea; siRNA; 59 ea), and blastocyst (PA, 98 ea; siRNA, 115 ea) stages. The embryo samples were isolated using TRIzol reagent (Sigma-Aldrich, USA), following the manufacturer’s protocol. The RNA samples were treated with DNase I to avoid genomic DNA contamination. The integrity of the RNA number (RIN > 7) was determined using an Agilent Technologies 2100 Bioanalyzer. The sequencing libraries were created from 50 ng of blastocyst samples and sequenced using the Ovation single-cell RNA-seq system. Raw RNA sequencing (RNA-Seq) reads were filtered, and the *Sus scrofa* reference genome (NCBI.Sscrofa10.2) was obtained, using Tophat (version 2.0.13) and Bowtie2 (version 2.2.3). Cufflinks (version 2.2.1) was used to assemble the transcript models from the alignments and estimate their abundance in the transcriptome. The transcript abundance was quantile normalized and also corrected for sequence bias to improve the expression estimates [21]. Differentially expressed genes (DEGs) were those that met the combined criteria of FDR-adjusted, *p* < 0.05, and absolute log_2_-fold change > 1, where a fold change was defined as expression in the samples of the siRNA-injected embryos divided by the expression in the control samples. The hierarchical cluster analysis was performed for the DEGs using MultiExperiment Viewer and Gene Ontology (GO). The pathway analysis was conducted using the online tool ToppCluster and the Database for Annotation, Visualization, and Integrated Discovery (DAVID).

### 2.8. RNA Isolation and Quantitative Reverse Transcription Polymerase Chain Reaction

The total RNA was isolated from 50 blastocysts using TRIzol reagent, and quantitative reverse transcription polymerase chain reaction (RT-qPCR) was performed using a Quantitative Real-Time RT-PCR Analysis Kit (Bio-Rad, Munich, Germany). The complementary DNA was synthesized using 1 µg of RNA, oligo (dT) primers, and the AMV First Strand cDNA Synthesis Kit (ROCHE). The quantitative PCR amplification was performed using SYBR^®^ Green EX Taq™ (TaKaRa) and an RG-6000 Real-Time PCR detection system (Corbett Research Co., Mortlake, Australia). The samples were run in triplicate. The relative gene expression was calculated using the comparative threshold cycle (Ct) method, with *GAPDH* as the reference gene. The thermocycling conditions were as follows: 95 °C for 10 min; followed by 35 cycles at 95 °C for 10 s, 60 °C for 30 s, and 72 °C for 30 s. Fluorescence was measured once. Primer sets for each gene are listed in Table 1.

### 2.9. Statistical Analysis

All data were analyzed using SAS Enterprise Guide 7.1 (SAS Institute Inc., Cary, NC, USA). The pre-implantation embryo development and gene expression levels were compared among all of the groups using ANOVA, followed by *t*-tests. The data are expressed as mean ± standard error of the mean (SEM). The significance was set at *p* < 0.05.

## 3. Results

### 3.1. Expression of cd26 mRNA in Porcine Pre-Implantation Embryos at Various Stages

We used RT-qPCR to measure the cd26 expression in the uninjected and cd26-siRNA-injected embryos in the following stages: MII; two-cell; four-cell; eight-cell; morula; and blastocyst (Figure 1). The expression of the cd26 mRNA in the two-cell and blastocyst embryos in the cd26 siRNA-injected groups was significantly lower than that in the parthenogenetic groups (*p* < 0.05).

### 3.2. Effect of cd26 Downregulation on the Development of Porcine Embryos

We first evaluated the dose sensitivity of cd26 to siRNA in porcine pre-implantation embryos. The injection of 10 and 20 μg siRNA significantly decreased the blastocyst development rate to 21.2% and 19.7%, compared with the uninjected group of 30.2% (Table 2). We selected 10 μg for further microinjection analyses. Our results showed that the developmental rates of early stage embryos (≤two-cell) did not differ between the cd26-siRNA-injected and uninjected or control-siRNA-injected embryos (Table 3). However, the siRNA-injected embryos developed into blastocysts at significantly lower rates (day 7: 13.9%) than the uninjected (43.4%) and control-siRNA-injected groups (39.6%) (*p* < 0.05; Table 3). Additionally, the total and apoptotic cell numbers significantly differed among blastocysts from the uninjected (46.3% and 3.8%), control-siRNA (49.6% and 4.5%), and cd26-siRNA groups (36.3% and 6.6%). The control embryos reached blastocyst stage on day 7, and expanded or hatching blastocysts were visible (data not shown). However, only a few cd26-siRNA-injected embryos reached the blastocyst stage during the same period.

### 3.3. Transcriptomic Analysis of Embryos upon cd26 Knockdown

Using RNA-Seq, we identified 38 DEGs (24 upregulated and 14 downregulated) in the day 7 cd26-siRNA-injected blastocysts (Figure 2a, Table 4 and Table 5). Highly upregulated genes included *RPLP2*, *EEF1B2*, *COX7C*, *RPS27A*, *RPL11*, *EEF1A1*, *HSPE1*, and *LOC100154750* (>five-fold). The downregulated genes were *GAN*, *RELL1*, *PTPRC*, and 11 novel transcripts. The GO analysis revealed significant enrichment of biological processes, such as translation (GO: 0006412), molecular function related to structural molecule activity and GTPase activity (GO: 0005198, GO: 0003924), and cellular components related to mitochondria (GO: 0005739). Additionally, the KEGG pathway associated with the ribosomes (SSC03010) was significantly upregulated (Figure 2b).

### 3.4. Validation of the Selected DEGs Using RT-qPCR

To validate the RNA-seq data, we performed RT-qPCR analysis of nine ribosomal genes (*RPLP2*, *RPL11*, *RPL12*, *RPS27A*, *RPS3A*, *RPS10*, *EEF1B2*, *EEF1A1*, and *NPC2*; Figure 3a), two ROS-related genes (*COX7C* and *CYCS*; Figure 3b), and two apoptosis-related genes (*HSPE1* and *CCNG1*; Figure 3c). These transcripts were selected based on their potential importance for the development of porcine embryos, as well as for the high log_2_FC value from the RNA-Seq data. We observed that the ribosomal protein-related genes were significantly increased in the cd26 siRNA-injected blastocysts, with the exception of the EEF1B2 gene. In the case of the EEF1B2 gene, the DEGs analysis showed a high value (7.25FC), but the RT-qPCR analysis did not show a significant difference. The ROS and apoptosis-related genes were significantly increased in the cd26 siRNA-injected blastocysts. These data were consistent with the results of the DEGs analysis.

## 4. Discussion

We previously reported that cd26 expression differs with the developmental stage and demonstrated the necessity of cd26 transcription for porcine embryo pre-implantation development. Downregulating the cd26 influenced the development to blastocyst formation, but not in the earlier stages. The development of the embryos showed no difference between the uninjected and con-siRNA groups in this study. Here, we used RNA-Seq to determine the role of cd26 between the uninjected and cd26-siRNA-injected groups. We identified several DEGs in the porcine blastocyst transcriptome of the cd26-siRNA-injection group. Among the most upregulated genes were the ribosomal protein-coding genes (*RPLP2*, *RPS27A*, *RPL11*, *RPS3A*, *RPL12*, and *RPS10*), eukaryotic translation factor-coding genes (*EEF1B2* and *EEF1A1*), reactive oxidative species-related genes, and apoptosis-related genes (*COX7C*, *CYCS*, *HSPE1*, and *CCNG1*). The most downregulated genes were uncharacterized with the exception of GAN, RELL1, and PTPRC. The role of the GAN gene is in the maintenance of the cytoskeletal or filamentary structure [22]. The RELL1 gene is known as one of the members of the tumor necrosis receptor family associated with embryo development [23]. It has been reported that the immuno-related gene PTPRC plays a role in implantation [24].

In terms of how these DEGs relate to the embryonic development, the ribosomal proteins are involved in protein translation, tumorigenesis, immune signaling, and development [25]. Dysfunctional ribosome biogenesis is associated with developmental defects [26,27]. The aberrant expression of *EEF1* regulates the epigenetic mechanisms at the chromatin level [28]. Additionally, oxidative damage and apoptosis are important factors in the IVM porcine embryos, as the excessive generation of reactive oxidative species is the main cause of oxidative damage and apoptosis in mammalian oocytes [29].

Overall, our findings suggest that failed gene transcript regulation causes developmental arrest after cd26 siRNA injection in porcine embryos. However, it was beyond the scope of our study to address the interaction between cd26, ribosomal proteins, oxidative damage, and apoptosis in porcine embryos. We believe that our study makes a significant contribution to the literature, because we provided the first empirical evaluation of cd26-regulated gene expression in porcine development. We provide important clarifications of the molecular mechanisms underlying the action of cd26 and useful information to improve the efficiency and success rate of in vitro porcine embryo production. Further experiments are needed to better understand the molecular mechanisms, including the cd26 signaling pathways, involved in successful pre-implantation embryonic development.

## 5. Conclusions

We demonstrated that cd26 knockdown negatively affected porcine parthenogenetic embryo development after siRNA injection into MII oocytes. We confirmed aberrant gene expression via the transcriptomic analysis of cd26-siRNA-injected and non-injected embryos, suggesting that cd26 has an important regulatory function in porcine embryo development. Thus, cd26 is a viable target for research aimed at improving the efficiency of IVC porcine embryos.

## Figures and Tables

**Figure 1 animals-12-01662-f001:**
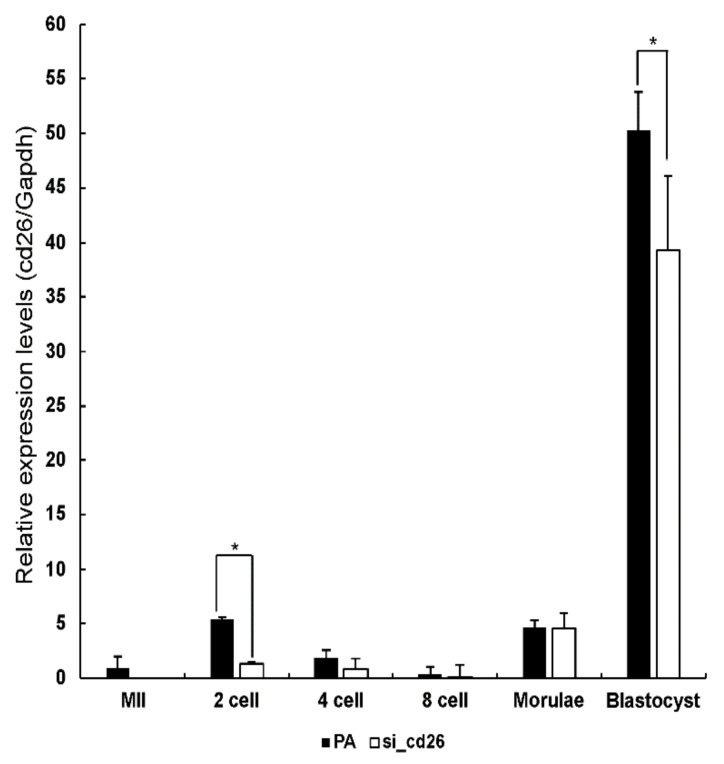
Relative abundance of cd26 transcripts in uninjected (PA) and siRNA-injected (si_cd26) porcine embryos. MII oocyte (*n* = 250), two-cell (parthenogenetically activated (PA), *n* = 200; siRNA, *n* = 195), four-cell (PA, *n* = 151; siRNA, *n* = 162), eight-cell (PA, *n* = 155; siRNA, *n* = 172), morula (PA, *n* = 50; siRNA, *n* = 59), and blastocyst stages (PA, *n* = 98; siRNA, *n* = 115). * *p* < 0.05, from the one-way ANOVA, followed by *t*-test (*p* < 0.05). Data are expressed as mean ± SEM.

**Figure 2 animals-12-01662-f002:**
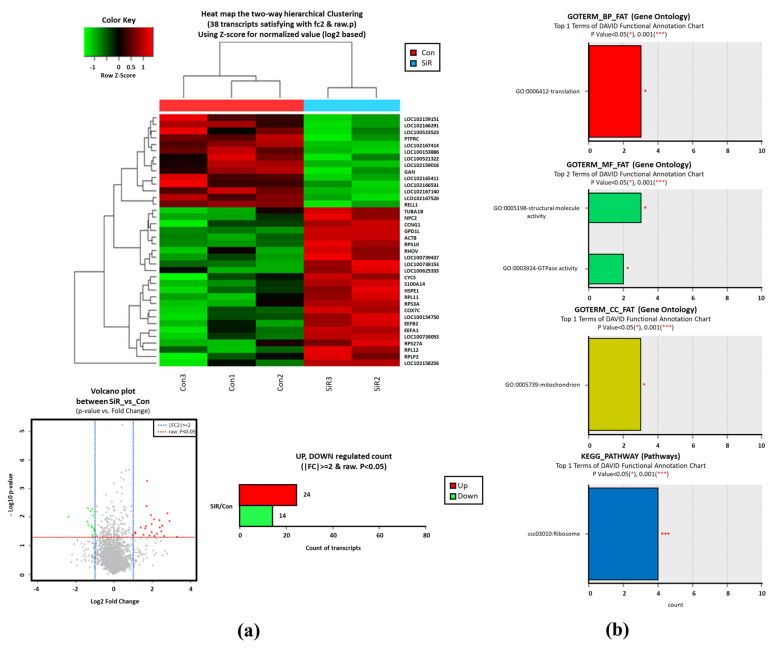
Hierarchical cluster, volcano plots and Gene ontology (GO) analysis of differentially expressed genes (DEGs; *P*-adjusted < 0.05 and log_2_ fold change) between uninjected parthenogenetic blastocysts (con.) and cd26-siRNA-injected parthenogenetic blastocysts (SiR.). Red and green blocks represent upregulated and downregulated genes, respectively; the color scale of the heatmap represents expression levels, where the brightest green stands for −1.0 log_2_ fold change and the brightest red stands for 1.0 log_2_ fold change. The 38 DEGs were selected based on the least standard deviation within a group (**a**). Functional analyses (GO and KEGG pathway) showed that DEGs were enriched in certain biological processes (BP), molecular functions (MF), and cellular components (CC), as well as ribosome-linked pathways (**b**).

**Figure 3 animals-12-01662-f003:**
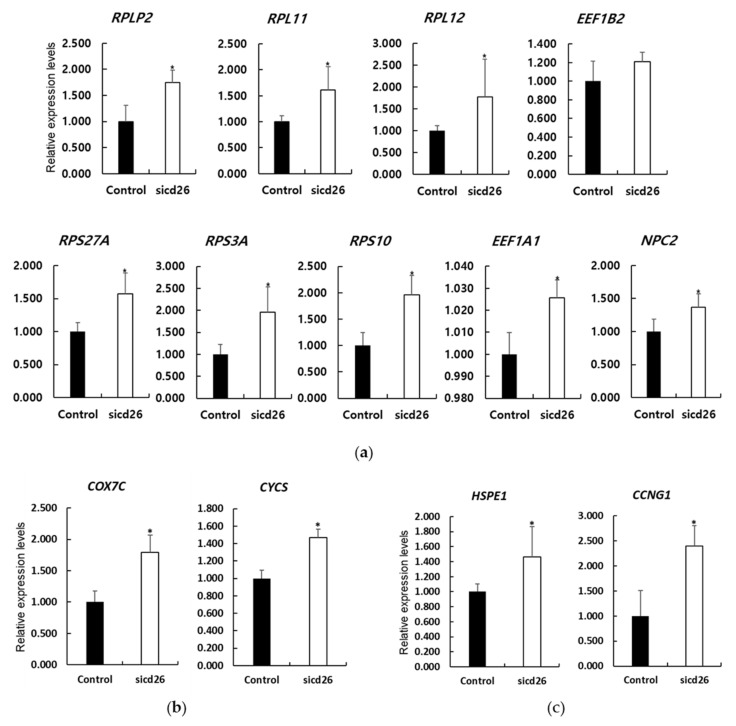
Comparison of mRNA expression levels of ribosomal protein (**a**); reactive oxygen species-(**b**); and apoptosis-related (**c**) genes in the control groups and cd26-siRNA-injected embryos. Bars represent mean ± SEM of three independent experiments; the results were analyzed using the one-way ANOVA. * *p* < 0.05 versus control.

**Table 1 animals-12-01662-t001:** Primers used for RT-qPCR.

Primer Set	Primer Sequence	Size (bp)	Gene Bank Accession No.
ACTB	F: 5′-CATCGCCGACAGGATGCAGAAG-3′	138	XM_021086047.1
	R: 5′-TGCTGGAAGGTGGACAGCGAG-3′		
RPLP2	F: 5′-GGGGCAATACTTCTCCTAGCG-3′	127	NM_001244866.1
	R: 5′-ATGACGTCCTCGATGTTCTTCC-3′		
EEF1B2	F: 5′-AAGAGGCTAAGAGAAGAACGCC-3′	167	NM_001243524.1
	R: 5′-ACCAAGCCATCTGCTTGAATGC-3′		
RPS27A	F: 5′-ATTGAGACTTCGTGGTGGTGC-3′	163	XM_003125136.5
	R: 5′-ATCTGAAGGGCACTCCCGAC-3′		
RPL11	F: 5′-GTCAGATCCTTTGGCATCAGGAG-3′	182	NM_001001638.1
	R: 5′-ATCCCCAGATCGATGTGTTCTTG-3′		
EEF1A1	F: 5′-TCTGGGAAAAAGCTGGAAGATGG-3′	161	NM_001097418.2
	R: 5′-CCCACAGCAACTGTCTGTCTC-3′		
RPS3A	F: 5′-AGGGTCGTGTGTTTGAAGTGAG-3′	147	NM_001137619.1
	R: 5′-AGCACATCTTGTCACGGGTAAG-3′		
RPL12	F: 5′-CCTCTGCCCTGATCATCAAAGC-3′	193	XM_005660442.3
	R: 5′-CATCCCACAGACTGAGCAGTCC-3′		
RPS10	F: 5′-ATGCTGATGCCCAAGAAGAACC-3′	126	NM_001244106.1
	R: 5′-ATTAGGCACATTCTTGTCCGCC-3′		
NPC2	F: 5′-AATCAACTGCCCCATCCAGAAAG-3′	134	NM_214206.1
	R:5′- AGCAGAAGAGACACTGGTCATTG-3′		
COX7C	F: 5′-CACAACCTCTGTGGTCCGTAG-3′	131	NM_001097474.1
	R: 5′-AAGGTGCAGCAAATCCAGATCC-3′		
CYCS	F: 5′-GGTCCAAACCTCCATGGTCTC-3′	110	NM_001129970.1
	R: 5′-ATCAGTGTCTCCTCTCCCCAG-3′		
HSPE1	F: 5′-GCTGAAACGGTAACCAAAGGAGG-3′	171	NM_214307.1
	R: 5′-GGTGCCTCCATATTCTGGCAG-3′		
CCNG1	F: 5′-TGCATTGGAGATCCAAGCACTG-3′	199	NM_001031781.2
	R: 5′-TGCAGTACGCCCAGAAACAATC-3′		
CD26	F: 5′-AAAGGCACCTGGGAAGTCATCG-3′	153	NM_214257.1
	R: 5′-CAGCTCACAACTGAGGCATGTC-3′		

**Table 2 animals-12-01662-t002:** Effect of cd26 siRNA concentration during in vitro culture on embryonic development.

Treatments	IVC	No. (%) of Embryos Developed to ^1^	
		2-Cell ≤ (%)	Total BL (%)
Control	106	84 (78.0 ± 5.4)	32 (30.2 ± 2.1)
5 μg siRNA	138	105 (83.3 ± 1.9)	35 (25.4 ± 3.3)
10 μg siRNA	132	98 (75.1 ± 3.6)	28 (21.2 ± 1.1) *
20 μg siRNA	137	104 (75.8 ± 2.3)	27(19.7 ± 1.6) *

^1^ All embryos were cultured until day 7. * *p* < 0.05, from the one-way analysis of variance, followed by *t*-tests. Data are expressed as mean ± SEM.

**Table 3 animals-12-01662-t003:** Development of porcine parthenogenetic embryos after injection with cd26 siRNA in the oocytes.

Treatments	IVC	No. (%) of Embryos Developed to ^1^		No. of Total Cells	Apoptotic Cells (%)
		≤Two-Cell (%)	Total BL (%)		
Uninjected	200	169 (84.5 ± 0.9)	84 (43.4 ± 3.8)	46.3 ± 2.9	1.8 (3.8 ± 0.5)
Con.-siRNA	171	144 (83.7 ± 0.9)	61 (39.6 ± 4.6)	49.6 ± 4.6	1.9 (4.5 ± 0.8)
Cd26 siRNA	152	131 (84.9 ± 1.2)	17 (13.9 ± 4.0) *	36.3 ± 2.7 *	2.3 (6.6 ± 0.6) *

^1^ All embryos were cultured until day 7. * *p* < 0.05, from the one-way analysis of variance, followed by *t*-tests. Data are expressed as mean ± SEM. IVC, in vitro culture; Con., control; BL, blastocyst; no. of total cells, total cell count in blastocysts.

**Table 4 animals-12-01662-t004:** Gene symbol, gene description, and logarithm of fold change (FC) of differentially expressed (upregulated) genes between cd26-siRNA-injected and non-injected porcine blastocysts.

Gene Symbol	Gene Description	FC
RPLP2	Ribosomal protein large P2	9.93
EEF1B2	Eukaryotic translation elongation factor 1 beta 2	7.52
COX7C	Cytochrome c oxidase subunit VIIc	7.00
RPS27A	Ribosomal protein S27a	6.23
RPL11	Ribosomal protein L11	5.83
EEF1A1	Eukaryotic translation elongation factor 1 alpha 1	5.64
HSPE1	Heat shock 10kda protein 1	5.33
LOC100154750	Nascent polypeptide-associated complex subunit alpha-like transcript variant x1	5.19
LOC102158256	Basic proline-rich protein-like	4.77
RPS3A	Ribosomal protein S3A	4.36
RPL12	Ribosomal protein L12	4.32
LOC100739053	Translationally controlled tumor protein pseudogene	3.96
RPS10	Ribosomal protein S10 transcript variant X1	3.93
S100A14	S100 calcium binding protein A14	3.84
CYCS	Cytochrome c somatic	3.65
GPD1L	Glycerol-3-phosphate dehydrogenase 1-like	3.38
ACTB	Actin beta	3.31
LOC100625333	High mobility group protein B2-like	3.22
NPC2	Niemann–Pick disease type C2	3.09
TUBA1B	Tubulin alpha 1b	2.95
CCNG1	Cyclin G1	2.66
LOC100739437	60S ribosomal protein L27a-like	2.19
RHOV	Ras homolog family member V	2.15
LOC100739153	Serine/arginine-rich splicing factor 3-like	2.01

**Table 5 animals-12-01662-t005:** Gene symbol, gene description, and logarithm of fold change (FC) of differentially expressed (downregulated) genes between cd26-siRNA-injected and non-injected porcine blastocysts.

Gene Symbol	Gene Description	FC
LOC102167140	T−cell receptor alpha chain V region CTL−L17-like	−5.22
LOC102165411	Uncharacterized LOC10216541 transcript variant X2	−2.61
LOC100153886	Olfactory receptor 11H6-like	−2.57
GAN	Gigaxonin	−2.54
LOC102167414	Putative uncharacterized protein DDB_G0271982-like	−2.31
LOC102159151	Uncharacterized LOC 102159151	−2.26
LOC102159016	Uncharacterized LOC 102159016	−2.21
LOC100523525	Claudin−1-like	−2.19
LOC102167526	Early endosome antigen 1-like transcript variant x3	−2.18
LOC102166291	Uncharacterized LOC 102166291	−2.17
RELL1	RELT-like 1	−2.14
LOC102166531	Uncharacterized LOC 102166531	−2.07
LOC100521322	Uncharacterized LOC 100521322	−2.04
PTPRC	Protein tyrosine phosphatase receptor type C	−2.04

## Data Availability

The data presented in this study are available on request from the corresponding author.

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
