# Peer review of "cd26 Knockdown Negatively Affects Porcine Parthenogenetic Preimplantation Embryo Development"

_animals, 2022, doi:10.3390/ani12131662_

Round 1

Reviewer 1 Report

In the current study, the reviewer thinks that using three groups (uninjected, Con-siRNA and Cd26 siRNA groups) is reasonable. However, in Figures 2 and 3, only two groups were used, why? In addition, what does control stand for? If control stands for uninjected group, why loss Con-siRNA group? If control stands for Con-siRNA group, why loss uninjected group? You need to explain it carefully. Discussion section needs to be improved carefully.

1)     TUNEL analysis section, there was no negative control statement.

2)     Table 1, please supplement primer length information.

3)     Lines 156-157, page 4, the sentence of “The expression of cd26 mRNA in two-cell and blastocyst embryos was significantly lower than that in the parthenogenetic groups” is unclear, please check it carefully again and re-write it.

4)     Figure 1, what does “PA” stand for? It is uninjected group?

5)     Lines 164-167, please delete this sentence.

6)     Line 178, page 6, you said that “compared with the control … …”. What does control stand for? Uninjected group or Con-siRNA group? Please explain it. This is a big problem, because in following results (e.g. Figures 2, 3), you lost one experimental group, uninjected group or Con-siRNA group.

7)     Lines 186-187, page 6, you said that “The control embryos reached blastocyst stage on day 7, and expanded or hatching blastocysts were visible”, but there were no data or pictures showed in your manuscript.

8)     3.4. Validation of the selected DEGs using RT-qPCR section, there was no detailed results presentation here.

9)     Discussion section needs to be strengthened carefully. You are required to discuss more your findings compared to other studies.

Author Response

In the current study, the reviewer thinks that using three groups (uninjected, Con-siRNA and Cd26 siRNA groups) is reasonable. However, in Figures 2 and 3, only two groups were used, why? In addition, what does control stand for? If control stands for uninjected group, why loss Con-siRNA group? If control stands for Con-siRNA group, why loss uninjected group? You need to explain it carefully. Discussion section needs to be improved carefully.

1)     TUNEL analysis section, there was no negative control statement.

Response: Thank you for your comment. we have added the negative control in the TUNEL analysis section. The sentence reads as follows;

“ The negative control embryos were placed in a well contatining FITC label only, without the TUNEL enzyme.”

2)     Table 1, please supplement primer length information.

Response: We have added the primer length in Table 1.

3)     Lines 156-157, page 4, the sentence of “The expression of cd26 mRNA in two-cell and blastocyst embryos was significantly lower than that in the parthenogenetic groups” is unclear, please check it carefully again and re-write it.

Response: We have changed the sentense. The sentence reads as follows;

“ The expression of cd26 mRNA in two-cell and blastocyst embryos in the cd26 siRNA injected groups was significantly lower than that in the parthenogetic groups”

4)     Figure 1, what does “PA” stand for? It is uninjected group?

Response: We are sorry for the confusion. We have changed the sentense. The sentence reads as follows;

“Figure 1. Relative abundance of cd26 transcripts in uninjected(PA) and siRNA-injected(si_cd26) porcine embryos.”

5)     Lines 164-167, please delete this sentence.

Response: Thank you very much for your comment. This is our mistake. This sentence is delete.

6)     Line 178, page 6, you said that “compared with the control … …”. What does control stand for? Uninjected group or Con-siRNA group? Please explain it. This is a big problem, because in following results (e.g. Figures 2, 3), you lost one experimental group, uninjected group or Con-siRNA group.

Response: Thank you very much for your comment. We are changed the sentence.

Line 178 “...compared with the unjected group(PA) of 30.2%.”

Line 181 “..the cd26-siRNA-injected and uninjected or control-siRNA-injected embryos.”

7)     Lines 186-187, page 6, you said that “The control embryos reached blastocyst stage on day 7, and expanded or hatching blastocysts were visible”, but there were no data or pictures showed in your manuscript.

Response: We have changed the sentence. The sentence reads as follows;

“The control embryos reached blastocyst stage on day 7, and expanded or hatching blastocysts were visible (data not shown).

8)     3.4. Validation of the selected DEGs using RT-qPCR section, there was no detailed results presentation here.

Response: Thank you for your comment. we have added the sentence in results section. The sentence reads as follows;

“ We observed that the ribosomal protein related genes were significantly increased in the cd26 siRNA injected blastocysts with the exception of EEF1B2 gene. In the case of the EEF1B2 gene, the DEGs analysis showed a high value(7.52 FC), but the RT-qPCR analysis not showed a significant difference. The ros- and apoptosis related genes were significantly increased in the cd26 siRNA injected blastocysts. These data were consistent with the results of DEGs analysis.”

9)     Discussion section needs to be strengthened carefully. You are required to discuss more your findings compared to other studies.

Response: We have added the sentence in discuss section. The sentence reads as follows;

“ We believe that our study makes a significant contribution to the literature because we provided the first empirical evaluation of cd26-regulated gene expression in porcine development. We provide important clarifications of the molecular mechanisms underlying cd26 action and useful information to improve the efficiency and success rate of in vitro porcine embryo production.”

We are sincerely appreciated for your kind help, time and affection for critical and helpful comment and suggestions in advance.

Reviewer 2 Report

The article has a good ideal to analysis differentially expressed genes in cd26 Knockdown and control by using transcriptome. However, there are some questions about this article.

1.     There are sloppy grammatical and editorial issues everywhere in this paper. Many of them could cause difficulties to the readers. Such as Line 16,19,23 and 26. The language is poor and carefully needs revision and polish. Please pay attention to the logic and accuracy of the language.

2.     The Materials and Methods should be written in the order in which the experiment was conducted, suggested 2.5 and 2.6 be transposed.

3.     There is a method of TUNEL test in the method, but it is not shown in the results.

4.     Suggest added 4 cell, 8 cell, Morulae and Blastocyst Effect of cd26 siRNA injectection on in vitro development of porcine embryos in table 2.

5.     Figure 2b. you should put it together.

6.     It is suggested to describe the sequencing results in detail and supplement the volcano map of differential expressed genes.

7.     It is suggested to added downregulated to verify result of RNA-sequencing.

8.     It is not clear in the method which period samples were RNA-sequencing.

9.     The discussion section did not elaborate on the expression of differential expressed genes after CD26 knockdown, whether the results of other studies in this area are consistent, and what effect it has on parthenogenesis.

Author Response

Comments and Suggestions for Authors

The article has a good ideal to analysis differentially expressed genes in cd26 Knockdown and control by using transcriptome. However, there are some questions about this article.

  1. There are sloppy grammatical and editorial issues everywhere in this paper. Many of them could cause difficulties to the readers. Such as Line 16,19,23 and 26. The language is poor and carefully needs revision and polish. Please pay attention to the logic and accuracy of the language.

Response: We commissioned a specialized institution to correct our paper.

  1. The Materials and Methods should be written in the order in which the experiment was conducted, suggested 2.5 and 2.6 be transposed.

Response: Thank you very much for your comment. We changed 2.5 and 2.6. The sentence reads as follows;

“2.5 TUNEL Aanalysis, 2.6. RNA extraction, cDNA library construction and RNA sequencing”

  1. There is a method of TUNEL test in the method, but it is not shown in the results.

Response: There are results on page 6, Line 179-186. 

  1. Suggest added 4 cell, 8 cell, Morulae and Blastocyst Effect of cd26 siRNA injectection on in vitro development of porcine embryos in table 2.

Response: We are checked the number of cleavage and blastocyst stage on days 2 and 7.

  1. Figure 2b. you should put it together.

Response: Figure 2b was put togeter.

  1. It is suggested to describe the sequencing results in detail and supplement the volcano map of differential expressed genes.

Response: Thank you for your comment. we have added the sentence and the volacno map in results section. The sentence reads as follows;

“ We observed that the ribosomal protein related genes were significantly increased in the cd26 siRNA injected blastocysts with the exception of EEF1B2 gene. In the case of the EEF1B2 gene, the DEGs analysis showed a high value(7.52 FC), but the RT-qPCR analysis not showed a significant difference. The ros- and apoptosis related genes were significantly increased in the cd26 siRNA injected blastocysts. These data were consistent with the results of DEGs analysis.”

  1. It is suggested to added downregulated to verify result of RNA-sequencing.

Response: We added the contents of downregulated genes. The sentece reads as follows;

“ The most downregulated genes were uncharacterized with the exception of GAN, RELL1 and PTPRC. The role of GAN gene is in the maintenance of cytoskeletal or filamental structure [21]. In RELL1 gene is  known as one of the members of the tumor necrosis receptor family associated with embryo development [22]. It has been reported that the immuno-related gene PTPRC a plays a role in implantation [23].     

  1. It is not clear in the method which period samples were RNA-sequencing.

Response: Thank you for your comment. We changed the sentence in this section. The sentence reads as follows;

“Sequencing libraries were created from 50ng of blastocyst samples and sequenced........” 

  1. The discussion section did not elaborate on the expression of differential expressed genes after CD26 knockdown, whether the results of other studies in this area are consistent, and what effect it has on parthenogenesis.

Response: We have added the sentence in discuss section. The sentence reads as follows;

“ We believe that our study makes a significant contribution to the literature because we provided the first empirical evaluation of cd26-regulated gene expression in porcine development. We provide important clarifications of the molecular mechanisms underlying cd26 action and useful information to improve the efficiency and success rate of in vitro porcine embryo production.”

We are sincerely appreciated for your kind help, time and affection for critical and helpful comment and suggestions in advance.

Reviewer 3 Report

The paper provides new data on gene related pig embryo development efficiency in vitro. Although it is interesting it needs some corrections before considering for publication.

IVP – is a generally knows abbreviation for in vitro (embryo) production, author use it in the text as in vitro produced, please change throughout the text to match common understanding of that abbreviation (=in vitro production)

Line 44-45: please reformulate it is not clear (insufficient cytoplasm…? Content?)

Line 17, 94: embryos? Or oocytes?

2.3. please add the number of oocytes injected in each group

Figure 1: the first sentence is not clear, injections were done into oocytes before parthenogenetic activation? Please check the same throughout the text

Please add section Experimental design – containing information on experimental total oocyte number, number of replicates, developmental stage and treatment in each group. Information from lines 154-155, 164-165 can be included there not in Results section

Results: the 30% of blastocyst in control group seems quite high for porcine embryos, please comment on it in discussion.

3.2 title: effect of cd26 down regulation in oocytes? (matured oocytes?) on….

Table 2: please develop titile, siRNA concentration in ……

Table 3: ..siRNA injection into…..

176-188: move to discussion

Figure 2, 3 and Table 4, 5: injection was done into oocyte or blastocyst? Please make clear in descriptions

3.4 validation – please move to MM section

Discussion – needs to be widen and deeper, please comment/discuss all the obtained results, not only the main findings. Furthermore, if I understand the experimental design correctly the cd26 knockdown was performed at oocyte level which then affected embryo development, and here mainly blastocyst formation…? If it is the case it should be mentioned and discussed. Same with the conclusions: the timing of procedure and then its further effects need to be mentioned.

Author Response

The paper provides new data on gene related pig embryo development efficiency in vitro. Although it is interesting it needs some corrections before considering for publication.

  1. IVP – is a generally knows abbreviation for in vitro (embryo) production, author use it in the text as in vitro produced, please change throughout the text to match common understanding of that abbreviation (=in vitro production)

   Response: We have changed the IVP abbrevation. The sentence reads as follows;

“ ...in vitro production (IVP)...”

  1. Line 44-45: please reformulate it is not clear (insufficient cytoplasm…? Content?)

Response: We changed to poor cytoplasm. Thank you for your comment.

  1. Line 17, 94: embryos? Or oocytes?

Response: Thank you for your comment. We are chagned the oocytes.

  1. 2.3. please add the number of oocytes injected in each group

Response: The number of oocytes used is shown in Table 2, Talbe 3 and Figure 1.

  1. Figure 1: the first sentence is not clear, injections were done into oocytes before parthenogenetic activation? Please check the same throughout the text

Response: You are correct. “ M&M 2.5, Line 96-98; The injected oocytes were activated electrically in a medium containing 0.3 M mannitol, 1.0 mM MgCl2, and 0.5 mM HEPES.”

  1. Results: the 30% of blastocyst in control group seems quite high for porcine embryos, please comment on it in discussion.

Response: In general, the blastocyst development rate of porcine parthenogenetic embryos is reported to be 30% to 50% [1. Hao J et al., (2021, Animals) : BL 49%, 2. Cao Z et al., (2020, Theriogenology) : BL about 30%, 3. Prather RS et al.,(2018, BOR): BL about 35%]. Therefore, I don’t think our data is very high. Thank you very much.

  1. 3.2 title: effect of cd26 down regulation in oocytes? (matured oocytes?) on….

Response: 3.2 title: Effect of cd26 downregulation on the development of porcine embryos. M&M 2.2: Matured oocytes used in microinjection experiments. The siRNA injected oocytes were activated electrically in a medium.

  1. Table 2: please develop titile, siRNA concentration in ……

Response: Thank you for your comment. we changed the sentence. The sentence reads as follows;

  “ Effect of cd26 siRNA concentration during in vitro culture on embryonic development”

  1. Table 3: ..siRNA injection into…..

Response: We changed the sentence. The sentence reads as follows;

“ Development of porcine parthenogenetic embryos after injection with cd26 siRNA in the oocytes.”

  1. 176-188: move to discussion

Response: We evaluated the dose sensitivity and the develpmental rates of cd26 to siRNA in porcine preimplantation embryos. these data describe the results. Therefore, these contents were considered correct to enter the field of results. Thak you for your sincere comment.

  1. Figure 2, 3 and Table 4, 5: injection was done into oocyte or blastocyst? Please make clear in descriptions

Response: We are sorry for the confusion. Injection was done into matured oocytes. We have added the sentense. The sentence reads as follows;

“M&M 2.5, Line113-114; Sequencing libraries were created from 50 ng of blastocysts samples and....”

  1. 3.4 validation – please move to MM section

Response: Thank you for your commnet. We identified 38 DEGs in day 7 blastocysts using RNA-sequencing and selected 9 ribosomal genes, 2 ros-related genes and 2 apoptosis-related genes for RT-qPCR. We observed that the ribosomal protein-, ros- and apoptosis-related genes were significantly increased in the cd26 siRNA injected blastocysts. Therefore, these data was considered correct to enter the field of results.

  1. Discussion – needs to be widen and deeper, please comment/discuss all the obtained results, not only the main findings. Furthermore, if I understand the experimental design correctly the cd26 knockdown was performed at oocyte level which then affected embryo development, and here mainly blastocyst formation…? If it is the case it should be mentioned and discussed. Same with the conclusions: the timing of procedure and then its further effects need to be mentioned.

Response: We have added the sentence in discuss section. The sentence reads as follows;

“ We believe that our study makes a significant contribution to the literature because we provided the first empirical evaluation of cd26-regulated gene expression in porcine development. We provide important clarifications of the molecular mechanisms underlying cd26 action and useful information to improve the efficiency and success rate of in vitro porcine embryo production.”

We are sincerely appreciated for your kind help, time and affection for critical and helpful comment and suggestions in advance.

Round 2

Reviewer 1 Report

In the last review, the reviewer suggested that using three groups (uninjected, Con-siRNA and Cd26 siRNA groups) was reasonable. In addition, the reviewer also suggested author explain why only use two groups (Control and Cd26 siRNA) in Figures 2 and 3, why lost Un-injected group? However, the authors deliberately avoided this problem. The reviewer thinks experimental design and results are unreasonable.

Author Response

In the last review, the reviewer suggested that using three groups (uninjected, Con-siRNA and Cd26 siRNA groups) was reasonable. In addition, the reviewer also suggested author explain why only use two groups (Control and Cd26 siRNA) in Figures 2 and 3, why lost Un-injected group? However, the authors deliberately avoided this problem. The reviewer thinks experimental design and results are unreasonable.

Response: Thank you for your comment. We are sorry for the confusion. We have changed the sentense. The sentence reads as follows;

“ Figure 2. Hierarchical cluster, volcano plots and Gene ontology(GO) analysis of differentially expressed genes (DEGs; p-adjusted<0.05 and log2 fold change) between uninjected parthenogenetic blastocysts(con.) and cd26-siRNA-injected parthenogenetic blastocysts(SiR).”

Our results showed that the cd26-siRNA-injected embryos developed at significantly lower rates (day 7: 13.9%) than the uninjected (43.4%) and control-siRNA-injected groups (39.6%) (p < 0.05, Table 3). Additionally, the total and apoptotic cell numbers significantly differed among blastocysts from the uninjected (46.3% and 3.8%), control-siRNA (49.6% and 4.5%), and cd26-siRNA groups (36.3% and 6.6%). The developmental rates of embryos did not differ between the uninjected and control-siRNA-injected embryos. Therefore, RNA-Seq. analysis was performed using only two groups (uninjected and cd26-siRNA-injected) in this study.

    we have added the sentence in discussion section(L234-236). The sentence reads as follows;

   “ The development of embryos was no difference between uninjected- and con-siRNA groups in this study. Here, we used the RNA-Seq to determine the role of cd26 between the uninjected and cd26-siRNA-injected groups.”

Again, we sincerely appreciate for your kind suggestion, comments and affection in advance.

Sincerely Yours

Mi-Ryung Park, PhD

Reviewer 2 Report

1. It is recommended to check the format of the full text

2. It is suggested to add RT-qPCR results of down-regulated genes to verify the accuracy of sequencing data

3. The picture combination needs to be beautified, figure 2b

Author Response

  1. It is recommended to check the format of the full text

Response: Thank you for your comment. We checked again according to the journal format.

  1. It is suggested to add RT-qPCR results of down-regulated genes to verify the accuracy of sequencing data

Response: Thank you very much for your careful consideration. We conducted the experiment by focusing only on the part of up-regulated genes. Because in the RNA-seq. analysis, 24 of the 38 gene differences were up-regulated genes. We haven’t considered the down-regulated genes. So, we’re going to think about the down-regulated(including unknown factors) genes in depth later.

  1. The picture combination needs to be beautified, figure 2b

Response: Thank you for your comment. we rearranged the picture (figure 2b).

Again, we sincerely appreciate for your kind suggestion, comments and affection in advance.

Sincerely Yours

Mi-Ryung Park, PhD

Reviewer 3 Report

Authors  corrected manuscript as suggested

Author Response

We sincerely appreciate for your kind suggestion, comments and affection in advance.